# BH4 activates CaMKK2 and rescues the cardiomyopathic phenotype in rodent models of diabetes

Hyoung Kyu Kim[1], Tae Hee Ko[1], In-Sung Song[1], Yu Jeong Jeong[1], Hye Jin Heo[1], Seung Hun Jeong[1], Min Kim[1], Nam Mi Park[1], Dae Yun Seo[1] , Pham Trong Kha[1], Sun-Woo Kim[1], Sung Ryul Lee[1], Sung Woo Cho[1,2], Jong Chul Won[1], Jae Boum Youm[1], Kyung Soo Ko[1], Byoung Doo Rhee[1], Nari Kim[1], Kyoung Im Cho[3], Ippei Shimizu[4], Tohru Minamino[4], Nam-Chul Ha[5], Young Shik Park[6], Bernd Nilius[7], Jin Han[1] 

**Diabetic cardiomyopathy (DCM) is a major cause of mortality/ morbidity in diabetes mellitus patients. Although tetrahydrobiopterin (BH4) shows therapeutic potential as an endogenous cardiovascular target, its effect on myocardial cells and mitochondria in DCM and the underlying mechanisms remain unknown. Here, we determined the involvement of BH4 deficiency in DCM and the therapeutic potential of BH4 supplementation in a rodent DCM model. We observed a decreased BH4:total biopterin ratio in heart and mitochondria accompanied by cardiac remodeling, lower cardiac contractility, and mitochondrial dysfunction. Prolonged BH4 supplementation improved cardiac function, corrected morphological abnormalities in cardiac muscle, and increased mitochondrial activity. Proteomics analysis revealed oxidative phosphorylation (OXPHOS) as the BH4-targeted biological pathway in diabetic hearts as well as BH4-mediated rescue of down-regulated peroxisome proliferator-activated receptor-γ coactivator 1-α (PGC-1α) signaling as a key modulator of OXPHOS and mitochondrial biogenesis. Mechanistically, BH4 bound to calcium/calmodulin-dependent protein kinase kinase 2 (CaMKK2) and activated downstream AMP-activated protein kinase/cAMP response element binding protein/PGC-1α signaling to rescue mitochondrial and cardiac dysfunction in DCM. These results suggest BH4 as a novel endogenous activator of CaMKK2.**

## Introduction

Diabetes mellitus (DM) is a metabolic disease with various complications, including diabetic cardiomyopathy (DCM), nephropathy, neuropathy, and encephalopathy. Among these, DCM is a major

cause of mortality and morbidity in DM patients (Boudina & Abel, 2010) and characterized by abnormal ventricle structure and function in diabetic patients without coronary artery disease or hypertension (Gilca et al, 2017). Patients or animal models of DCM exhibit structural cardiac remodeling, such as ventricular hypertrophy and interstitial fibrosis, and functional impairments, including systolic and diastolic dysfunctions (Gilca et al, 2017). Currently, there is no specific therapy for DCM patients in clinical practice, despite the critical need (Gilca et al, 2017).

Increased oxidative stress, alterations of energy metabolism, and apoptotic cardiac cell death resulting from mitochondrial dysfunction are implicated in DCM pathogenesis and represent potential therapeutic targets (Duncan, 2011). Studies of diabetic animal models reveal impairments in state-3 mitochondrial oxygen consumption, respiratory chain-complex activity, and mitochondrial ultrastructure and proliferation in the heart (Duncan, 2011). Similarly, patients with type 2 DM show abnormal ATP generation, fatty acid utilization, and oxidative phosphorylation (OXPHOS) in cardiac mitochondria (Anderson et al, 2009). However, few studies have demonstrated a beneficial effect of mitochondrion-targeted antioxidant therapy in DCM (Sharma, 2015).

Tetrahydrobiopterin (BH4) is a multifunctional cofactor implicated in the regulation of the nervous, immune, and cardiovascular systems and exhibits combined activities as an enzymatic cofactor, a cofactor for nitric oxide (NO) synthesis, and/or a scavenger of reactive oxygen species (ROS) (Kim et al, 2010). However, BH4 is also susceptible to oxidation, with reductions in BH4 levels reported in the presence of high levels of oxidative stress (Kim et al, 2010). Therefore, low levels of BH4 are associated with a broad range of cardiovascular diseases, including hypertension, hypertrophy, and ischemic heart disease, as well as DM (Kim et al, 2010; Arning et al, 2016). Notably, BH4 is a well-established cofactor for endothelial nitric oxide synthase (eNOS/NOS3), regulating vascular and cardiac

[1]Department of Physiology, BK21 Plus Project Team, College of Medicine, Smart Marine Therapeutics Center, Cardiovascular and Metabolic Disease Center, Inje University, Busan, Republic of Korea   [2]Division of Cardiology, Department of Internal Medicine, Inje University College of Medicine, Ilsan Paik Hospital, Goyang, Korea   [3]Division of Cardiology, Department of Internal Medicine, College of Medicine, Kosin University, Busan, Republic of Korea   [4]Department of Cardiovascular Biology and Medicine, Niigata University Graduate School of Medical and Dental Sciences, Niigata, Japan   [5]Department of Agricultural Biotechnology, Seoul National University, Seoul, Republic of Korea   [6]School of Biotechnology and Biomedical Science, Inje University, Kimhae, Republic of Korea   [7]Katholieke Universiteit Leuven, Department of Cellular and Molecular Medicine, Leuven, Belgium

Correspondence: phyhanj@inje.ac.kr

**Table 1.   Blood component analysis in DCM model rats.**

| Component | LETO | OLETF | OLETF/BH4 |
|---|---|---|---|
| CPK (U/l) | 89.6 ± 5.3 | 223.6 ± 12.4[a] | 129 ± 20.8[b] |
| LDH (U/l) | 349.5 ± 105.7 | 936 ± 155.8[a] | 332 ± 55.2[b] |
| Myoglobin (ng/ml) | 133.6 ± 48.2 | 70.2 ± 19.3 | 55.7 ± 26.6 |
| Low-density lipoprotein (mg/dl) | 16.1 ± 1.6 | 26.3 ± 3.6[a] | 41.7 ± 7.1[a] |
| High-density lipoprotein (mg/dl) | 25.4 ± 0.9 | 40 ± 2.9[a] | 48.6 ± 2.8[a] |
| Albumin (g/dl) | 3.2 ± 0.1 | 2.5 ± 0.1[a] | 2.4 ± 0.1[a] |
| Glucose (mg/dl) | 193.6 ± 8.7 | 500 ± 40.4[a] | 500.6 ± 0.6[a] |
| Total cholesterol (mg/dl) | 89.6 ± 5.1 | 147.6 ± 15[a] | 189.6 ± 17.4[a] |
| Blood urea nitrogen (mg/dl) | 16.2 ± 0.3 | 18.4 ± 2.1 | 18.3 ± 0.6 |
| Creatinine (mg/dl) | 0.46 ± 0.03 | 0.33 ± 0.08 | 0.33 ± 0.06 |
| Triglyceride (mg/dl) | 31 ± 5.1 | 189.6 ± 69.9[a] | 262 ± 35.5[a] |

[a]$P < 0.05$ versus LETO.
[b]$P < 0.05$ versus OLETF.

function. The BH4-eNOS uncoupling is associated with vascular disease, hypertrophic cardiac remodeling, and ischemia–reperfusion injury (Kim et al, 2010). We previously demonstrated that BH4 deficiency increases ROS generation and mitochondrial dysfunction independent of NO (Kim et al, 2007) and that a mouse model of BH4 deficiency (sepiapterin reductase-knockout [$Spr^{-/-}$] mice) displayed severe functional impairment in the heart and mitochondria, which was rescued by exogenous BH4 supplementation (Kim et al, 2019). Moreover, a recent study suggested that BH4 protects against hypertrophic heart disease independent of myocardial nitric oxide synthase (NOS) coupling (Hashimoto et al, 2016). However, the effect of BH4 on myocardial cells and mitochondria in DCM remains unknown.

This study aimed to determine the involvement of BH4 deficiency in DCM and the ability of BH4 supplementation to restore mitochondrial and heart function during late-stage DCM in rat models (Otsuka Long–Evans Tokushima Fatty [OLETF] rats [Hayashi et al, 2003], $db/db$ mice [Verma et al, 2018], and $Spr^{-/-}$ mice Kim et al, 2019]).

# Results

## DCM is associated with BH4 deficiency, which is improved by BH4 supplementation

In OLETF rats fed a normal diet, diabetes develops at ~28 wk and cardiac dysfunction at 50 wk, which was confirmed here by fasting glucose levels, i.p. glucose-tolerance tests, and higher levels of various serum markers at 50 wk relative to those in Long–Evans Tokushima Otsuka (LETO) controls (Table 1). BH4 supplementation had no effect on body weight or blood glucose levels in OLETF rats (Fig 1B and C). Interestingly, in the BH4 treatment group, low-density lipoprotein (LDL), total cholesterol (TC), and triglyceride (TG) levels tended to be higher than those in the OLETF group; however, this was not statistically significant (Fig S1).

Echocardiography performed to monitor left ventricle (LV) contractility in animals between 40 and 50 wk of age confirmed that systolic dysfunction emerged at 48 wk, the time at which BH4 supplementation was initiated (Fig 1A). Treatment with BH4 for 2 wk significantly improved fractional shortening in the hearts of OLETF rats (Fig 1D and E). Strain echocardiography is an effective tool for assessing regional and global modifications in LV function induced by cardiac fibrosis and hypertrophy (Haland et al, 2016). Here, we found that radial strain analysis of LV walls revealed significantly increased contraction in BH4-treated OLETF rats (Fig 1F and G).

To determine whether these effects were derived from cardiomyocytes, we measured electrically stimulated sarcomere shortening in isolated LV cardiomyocytes. We found that impaired contractility in cells from OLETF rats was reversed following BH4 supplementation (Fig 1H–J), and that a similar effect was observed in 20-wk-old $db/db$ model hearts via 2D M-mode echocardiography (Fig S2A–C), which revealed significantly improved cardiac contractility.

## BH4 induces reductions in histopathological signs of cardiomyopathy in DM model rats

BH4 supplementation reversed increases in the cross-sectional size of cardiomyocytes measured at post-papillary muscle and proximal coronary artery areas (Fig 2A and B). In addition, heart tissues from OLETF rats showed large amounts of fibrotic collagen deposition, which were reduced by BH4 supplementation (Fig 2C and D) and BH4-attenuated apoptotic cell death in OLETF hearts (Fig 2E and F). To elucidate the molecular mechanism of the anti-fibrotic and anti-hypertrophic effects induced by BH4 treatment, we tested proteasome activity, heart inflammation, and typical fibrosis signals associated with cardiac remodeling (Fig S3).

The measurement of the activities of matrix metalloproteinases (MMPs), which are elevated in heart-failure patients and by exercise-induced cardiac hypertrophy (Liu et al, 2006), revealed significant increases in proMMP2 and proMMP9 levels in OLETF hearts, with reversals in these levels observed after BH4 supplementation (Fig S3A and B). Similarly, we found that the activity of the 26S proteasome was significantly elevated in OLETF hearts and reversed following BH4 supplementation (Fig S3C). The specificity of

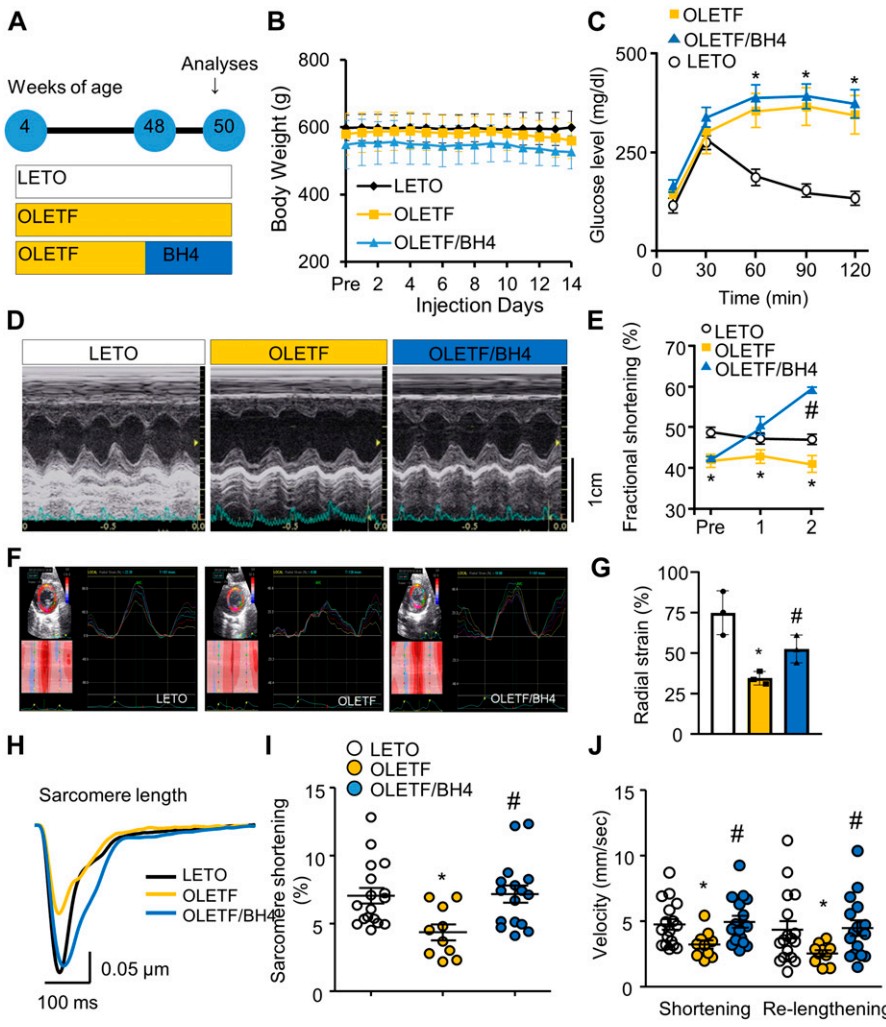

**Figure 1. BH4 supplementation recovers cardiac systolic dysfunction in a rat model of late-stage type 2 diabetes mellitus.**
**(A)** Experimental design. **(B, C)** Body weight (B) and i.p. glucose-tolerance tests (n = 5/group) (C). **(D)** Representative M-mode echocardiography at 50 wk. **(E)** Fractional shortening during pretreatment and after 1 and 2 wk of BH4 or vehicle treatment (n = 5/group). **(F)** 2D echocardiography results. **(G)** Comparison of radial strain (n = 5/group). **(H)** Average traces of electrical stimulation–induced sarcomere shortening in isolated left ventricle (LV) cardiomyocytes (n = 15 cells from three animals/group). **(I)** Peak shortening of isolated LV cardiomyocytes (n = 15/group). **(J)** Velocity of shortening and Re-lengthening of isolated LV cardiomyocytes (n = 15/group). *P < 0.05 versus Long–Evans Tokushima Otsuka; #P < 0.05 versus Otsuka Long–Evans Tokushima Fatty.

the 26S proteasome activity was confirmed using lactacystin, a 26S proteasome inhibitor.

Immunoblot analysis showed increased monocyte chemo-attractant protein-1 levels as a DCM biomarker (Dobaczewski & Frangogiannis, 2009) in cardiac tissues from OLETF rats, whereas levels of TGF-β1, collagens I and III, and fibronectin were similar to those in LETO rats, likely due to the old age of the animals. Interestingly, BH4 supplementation reduced active profibrogenic factors to levels lower than those in LETO rats (Fig S3D and E), demonstrating the strong antifibrotic capability of BH4. To confirm the change in eNOS-mediated NO signal by BH4 treatment, we tested the level of eNOS and phosphor-eNOS in the heart of LETO, OLETF, and OLET/BH4 rats. The ratio of p-eNOS/eNOS in the BH4 treatment group tended to increase, but there was no statistical significance (Fig S3F).

## Mitochondrial dysfunction in DM is associated with BH4 deficiency, which is improved by BH4 supplementation

Although total biopterin levels did not differ between LETO and OLETF rats, BH4:total biopterin ratios were significantly lower in the

hearts and mitochondria of OLETF rats (Fig 3A and B). Moreover, BH4 supplementation increased levels of total biopterin, restored altered BH4:total biopterin ratios, and significantly reduced levels of lactate dehydrogenase (LDH) and creatine phosphokinase (CPK) as known markers of cardiac damage (Fig 3C) (Hung et al, 2004). In addition, OLETF rats exhibited less ATP production (Fig 3D) than did LETO rats and depolarized inner mitochondrial membrane potentials (Fig 3E). Furthermore, mitochondrial damage resulted in elevated ROS levels in isolated mitochondria in the presence of a complex I inhibitor (rotenone) that mimics mitochondrial stress (Fig 3F). At the basal level, where there was only mitochondria without substrate, mitochondria produced very low levels of ROS. However, in the presence of a substrate and the complex I inhibitor, rotenone, OLETF cardiac mitochondria produced significantly high ROS levels compared with those in the others, which led to increased oxidative stress in cardiac tissue (Fig 3G).

In assays of OXPHOS-complex activity, complexes I, III, and V activities (Fig 3H–M) were reduced in the OLETF group. However, BH4 supplementation successfully rescued mitochondrial abnormalities and reduced oxidative stress in OLETF rats. Mitochondrial dysfunction is generally associated with structural deformation in

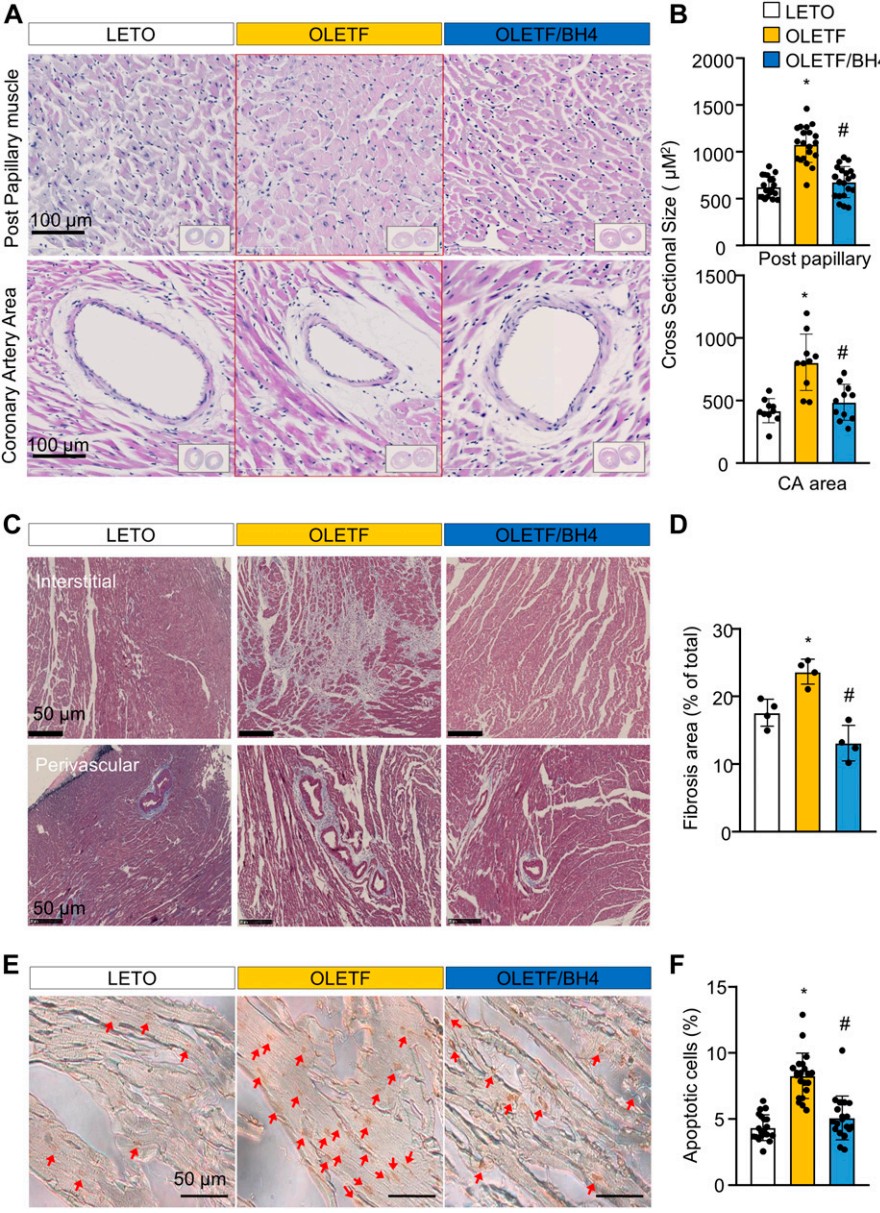

**Figure 2. BH4 supplementation recovers pathological cardiac remodeling in a model of late-stage type 2 diabetes mellitus.**
**(A)** Representative H&E staining of cross-sectioned hearts. **(B)** Cross-sectional size of cardiac muscle fibers. **(C)** Masson's trichrome staining of cross-sectioned hearts. **(D)** Quantification of fibrotic area. **(E)** Images of TUNEL-stained heart tissues from experimental rats. Red arrow: TUNEL+ apoptotic cells. Scale bar: 50 $\mu m$. **(F)** Quantification of apoptotic cell death. Data in (A, B, C, D, E, F) represent the mean ± SEM ($n = 4$ animals/group). *$P < 0.05$ versus Long–Evans Tokushima Otsuka; #$P < 0.05$ versus Otsuka Long–Evans Tokushima Fatty. TUNEL, terminal deoxynucleotidyl transferase dUTP nick-end labeling.

DCM models, and OLETF rats and *db/db* mice exhibited larger percentages of damaged mitochondria accompanied by disrupted matrices, all of which were normalized by BH4 supplementation (Fig S4A–D).

## BH4 supplementation alters OXPHOS-related protein levels

To elucidate the underlying mechanisms of BH4-induced cardiac recovery in DCM, we performed liquid chromatography tandem mass spectrometry (LC–MS/MS) analyses. We identified 878 cardiac proteins in LV tissues, among which 142 (16%) exhibited a twofold change (up or down) between OLETF/BH4 and OLETF groups, with these representing 142 differently expressed proteins (DEPs) in this study. Construction of a DEP-specific protein–protein interaction network resulted in a network containing 109 proteins and 364

interactions. Eighteen DEPs (red circles) were directly linked to four major cardiac regulatory pathways (red triangles), including "dilated cardiomyopathy," "adrenergic signal," "PPAR-signal," and "OXPHOS" (Fig 4A). 37 proteins were linked to the 18 hub proteins and represented second link proteins (purple circles). The remaining 50 proteins were linked to second link proteins and represented third link proteins (blue circles). Among the four biological pathways, the OXPHOS pathway showed the highest enrichment *P*-value, with eight component proteins (Fig 4B and C). These results suggested OXPHOS as the possible BH4 biological target pathway. In addition to the four major cardiac regulatory pathways, functional enrichment associated with "glutathione metabolism" or "proteasome" was also identified. These pathways appear to be associated with oxidative stress and pathological remodeling, respectively, in DCM (Fig S5).

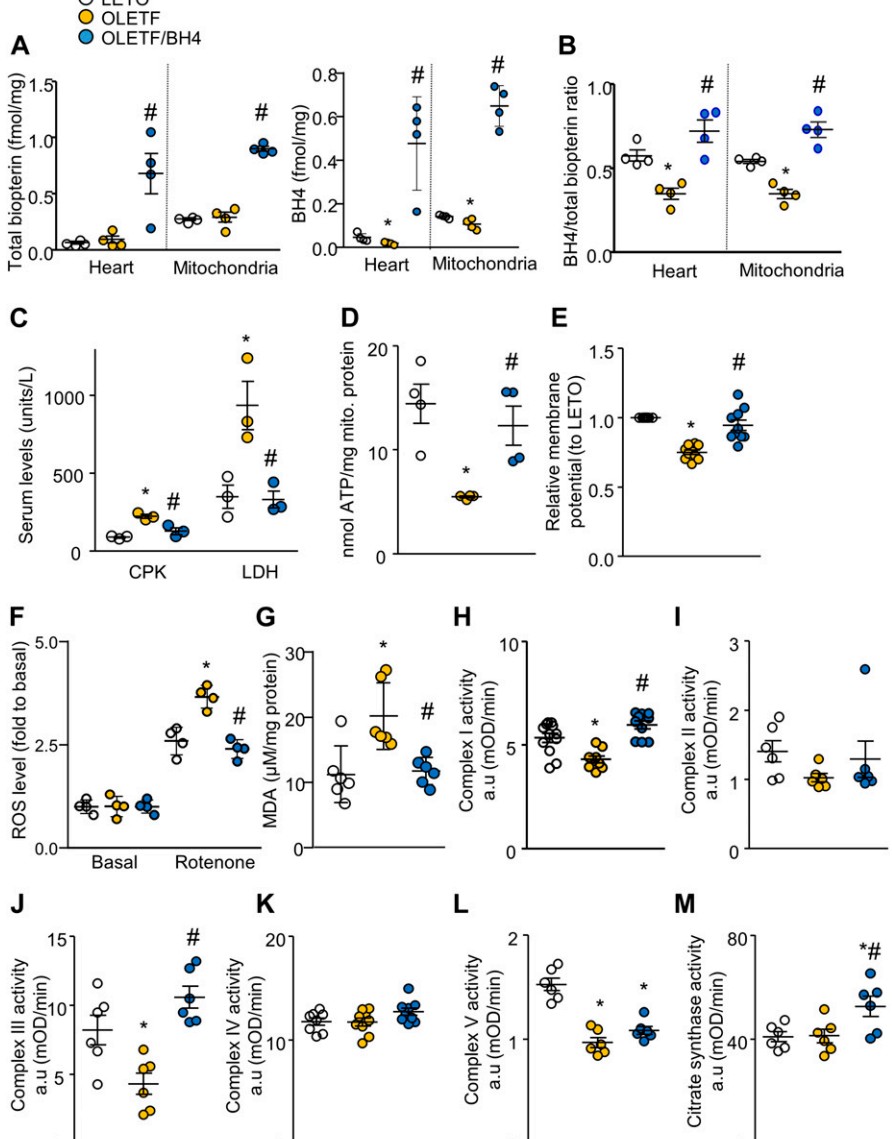

**Figure 3.  BH4 supplementation improves the biochemical phenotype of diabetic rat hearts.**
**(A)** Total biopterin and BH4 level in heart and mitochondria ($n = 4$/group). **(B)** BH4/total biopterin ratios ($n = 4$/group). **(C)** Serum levels of CPK and LDH. **(D)** ATP levels. **(E)** Mitochondrial membrane potential. **(F)** Reactive oxygen species production assessed by relative Amplex Red intensity. **(G)** Lipid oxidation assay to measure oxidative stress in heart tissue. **(H, I, J, K, L)** Activities of OXPHOS complexes I–V. **(M)** Citrate synthase activity. *$P < 0.05$ versus Long–Evans Tokushima Otsuka; #$P < 0.05$ versus Otsuka Long–Evans Tokushima Fatty.

We then evaluated changes in the levels of major OXPHOS proteins (NADH:ubiquinone oxidoreductase core subunit S8 [NDUFS8], succinate dehydrogenase complex iron sulfur subunit B [SDHB], ubiquinol-cytochrome C reductase core protein 2 [UQCRC2], and ATP synthase $\alpha$ subunit [ATP5A]) and master transcriptional regulators of mitochondrial biogenesis and OXPHOS (peroxisome proliferator-activated receptor-$\gamma$ coactivator 1-$\alpha$ [PGC-1$\alpha$], total and phosphorylated [p-] forms of AMP-activated protein kinase [AMPK]-$\alpha$ and $-\beta$, and phosphorylated forms of cAMP response element binding protein [p-CREB]) in LETO, OLETF, and OLETF/BH4 rats. OLETF rats showed lower levels of p-CREB, the upstream regulator of PGC-1$\alpha$ (Fig 4D), and NDUFB8, SDHB, UQCRC2, and ATP5A as representative mitochondrial complex-1, -2, -3, and -5 component proteins, respectively (Fig 4E). BH4 supplementation reversed these changes and enhanced levels of p-AMPK-$\alpha$, a functional regulator of PGC-1$\alpha$ (Fernandez-Marcos & Auwerx, 2011), and electron-transport chain-complex proteins,

suggesting AMPK/CREB/PGC-1$\alpha$ signaling as a possible BH4 target for the repair of mitochondrial dysfunction.

### BH4 activates calcium/calmodulin-dependent protein kinase (CaMK) kinase 2 (CaMKK2)–mediated CaMK type IV (CaMKIV)/CREB signaling

To identify the upstream regulators of PGC-1$\alpha$, we screened kinases according to in vitro activity (Fig S6) and ligand–protein binding simulation assays. To test the direct binding of BH4 and CaMKIV or CaMKK2, we performed a computational protein–ligand docking simulation and surface plasmon resonance (SPR) binding assay. Computational protein–ligand docking simulation results suggested a probable binding of BH4 to CaMKIV (Protein Data Bank: 2W4O) or indicated CaMKK2 (Protein Data Bank: 6CMJ) domain residues by polar, hydrophobic, and hydrogen bond interactions

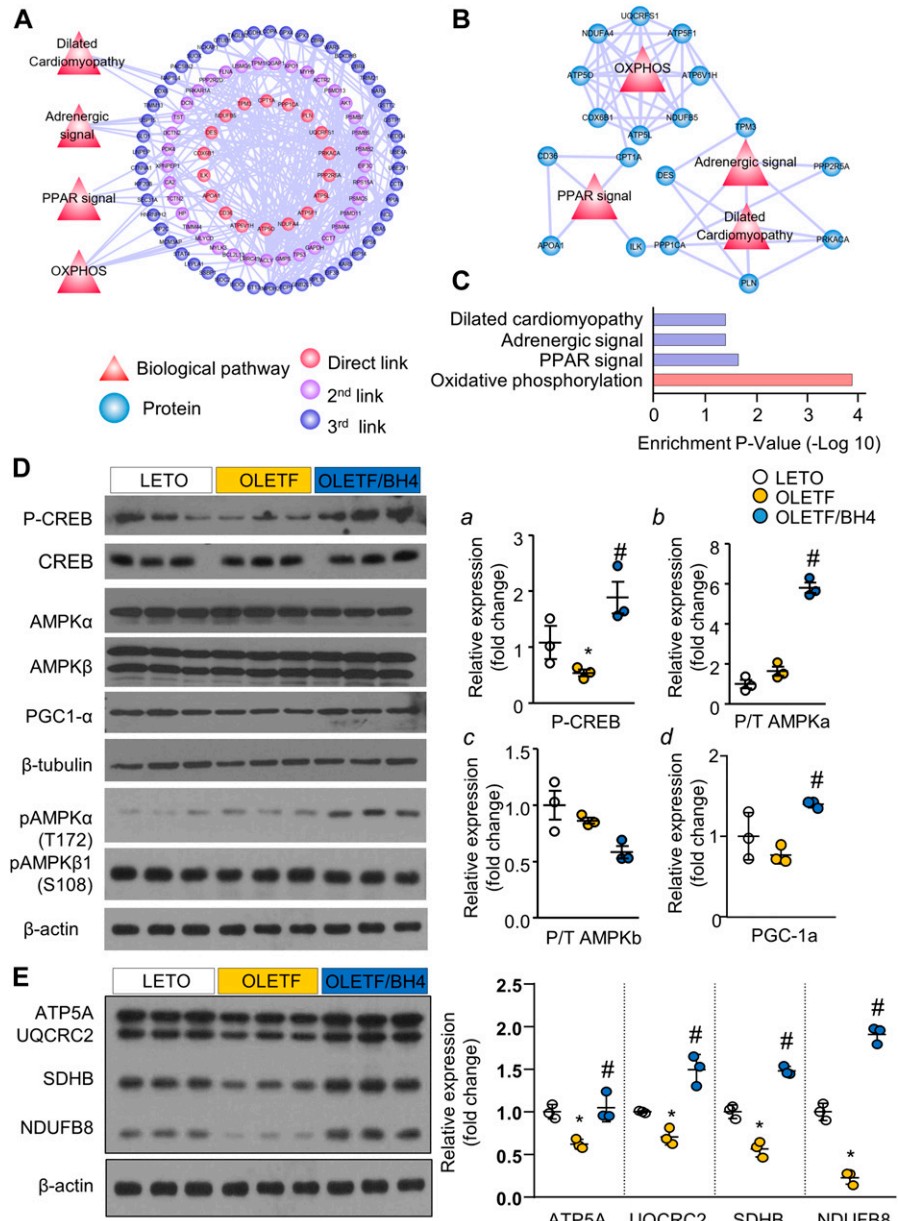

**Figure 4.   PGC-1α mediates the OXPHOS pathway and a primary BH4 target.**
**(A)** Differently expressed proteins between Otsuka Long–Evans Tokushima Fatty (OLETF) and OLETF/BH4 hearts and related cardio-relevant biological pathways. Red circles represent proteins directly linked to four biological pathways (direct links). Purple circles represent proteins that interact with red-circle proteins (second link). Blue circles represent proteins that interact with purple-circle proteins (third link). **(B)** Four biological pathways and their component proteins. **(C)** Enrichment *P*-values of four BH4 targets related to cardio-relevant biological pathways associated with heart function in the DCM model. **(D, E)** Immunoblot analysis of proteins related to mitochondrial biogenesis and functional regulation of (D) mitochondrial OXPHOS complexes (E) (*n* = 3/group). ATP5A for complex V, UQCRC2 for complex III, SDHB for complex II, and NDUFSB8 for complex I. Relative protein expression was normalized to β-actin. *P < 0.05 versus Long–Evans Tokushima Otsuka; #P < 0.05 versus OLETF. P/T, phosphorylated: unphosphorylated (total) protein ratio.
Source data are available for this figure.

(Fig 5A–D). The SPR assay was performed to validate the docking simulation results of BH4 binding to CaMKIV or CaMKK2, which phosphorylates AMPK (Racioppi & Means, 2012). The SPR analysis revealed that CaMKK2 directly and dose-dependently bound to BH4 (Fig 5F); however, CaMKIV was not the direct binding target of BH4 (Fig 5E). To confirm these results, we assessed levels of total and phosphorylated CaMKIV, CREB, p38 MAPK, and AMPK-*α* and -*β*1 via Western blot in *Spr⁻/⁻* mice. Compared with wild-type (WT) mice, BH4-deficient *Spr⁻/⁻* mice displayed significantly reduced phosphorylation of all these proteins, as well as reduced total levels of CREB, AMPK-*α* and -*β*1, and PGC-1*α* (Fig 6A and B). Furthermore, BH4 supplementation restored these levels to those observed in WT mice, except for levels of phosphorylated p38 MAPK.

In addition to its effect of restoring the pathological condition, the effect of BH4 in normal healthy cell lines, animal hearts, and mitochondria was also tested (Fig S7). To confirm this direct effect of BH4 on cardiac mitochondria, we analyzed the effect of BH4 on mitochondrial oxygen consumption rate in purely isolated mitochondria of mice. Final concentration of 20 *μ*M of BH4 was added into the oxygen consumption rate assay chamber together with isolated mitochondria without preincubation. After a stabilization period of 5 min, glutamate/malate and ADP were sequentially added to measure mitochondrial oxygen consumption. In the presence of BH4, mitochondrial oxygen consumption increased in both state 4 (glutamate and malate) and state 3 (ADP) (Fig S7A). The increase in mitochondrial biosynthesis by BH4 was confirmed in HL-1 cells (Fig S7B).

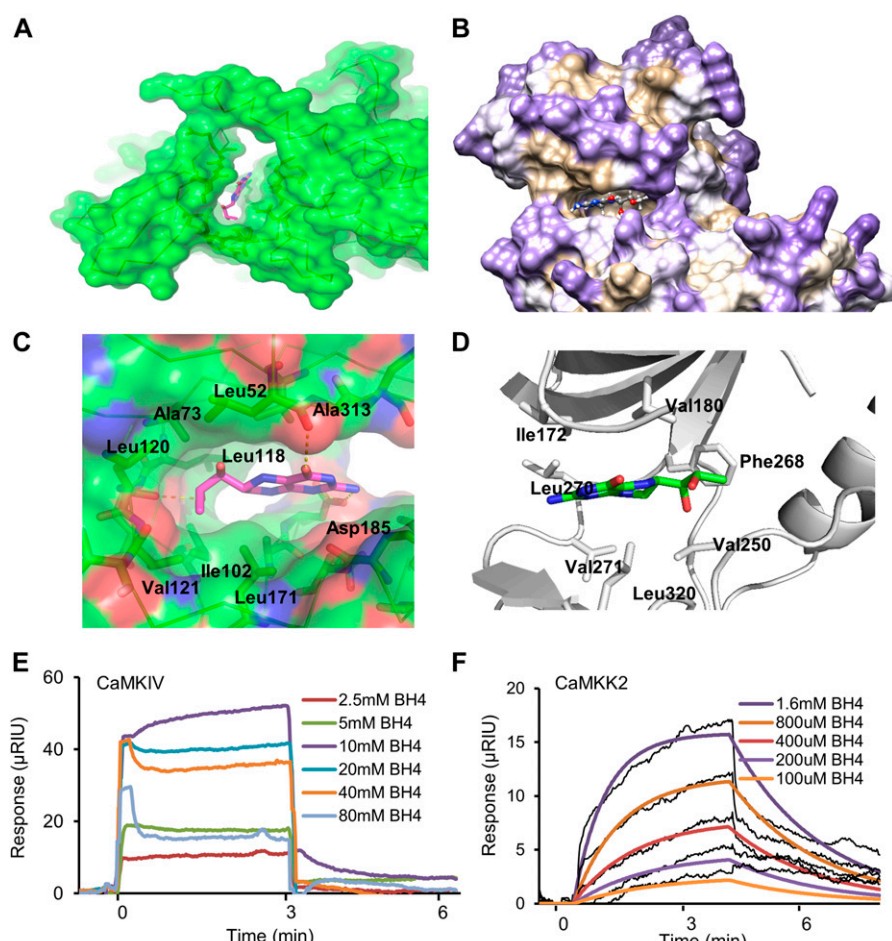

**Figure 5.  CaMKK2 is a novel BH4-binding target.**
**(A, B, C, D, E, F)** Protein–ligand docking simulation results for BH4–CaMKIV (A, B) and BH4–CaMKK2 (D, E). **(C, F)** Sensorgrams for BH4 binding to recombinant CaMKIV (C) or CaMKK2 (F).

The mitochondrial mass was measured by acridine orange 10-nonyl bromide (NAO; Invitrogen), which binds to the mitochondrial membrane phospholipid, cardiolipin. BH4 treatment also increased the protein level of PGC-1α, mt-TFA, and NRF1, which are key regulators of mitochondrial biogenesis, in treated WT mice (Fig S7C). These results suggest that BH4 can enhance mitochondrial function not only under pathological conditions but also under normal conditions in the heart.

We knocked down *Spr* in mouse cardiac HL-1 cells to test the effect of BH4 deficiency on mitochondrial functions without systemic effect. Depletion of the *Spr* gene depolarized mitochondrial inner-membrane potentials, reduced total cellular oxygen consumption, and decreased ATP levels in HL-1 cells (Fig 7A–C), all of which were rescued by BH4 supplementation. Importantly, *Spr* knockdown also reduced the protein level of total and phosphorylated CREB and AMPK-α, which was subsequently enhanced by BH4 supplementation (Fig 7D). In addition, *CaMKK2* knockdown in HL-1 cells via siRNA decreased the level of PGC-1α, -1β, and AMPK phosphorylation, which abolished BH4-induced increases in PGC-1α levels (Figs 7E and S8). BH4 supplementation did not directly alter Ca²⁺ transients or L-type Ca²⁺ currents in single cardiac myocyte, which excluded the possibility of Ca²⁺-mediated CaMKK2 activation in BH4 treatment (Fig S9). These results suggest that BH4

directly influences mitochondrial function in cardiac cells via interactions with CaMKK2 to activate CREB/AMPK/PGC-1α signaling.

## Discussion

DCM is a leading cause of death in DM patients. Although drugs that lower blood glucose levels and/or affect hyperlipidemia show beneficial effects against DCM, there remains no established treatment modality (Miki et al, 2013). As a novel therapeutic target for diabetes and its complications, BH4 supplementation improves insulin sensitivity (Nystrom et al, 2004) and attenuates hyperglycemia (Abudukadier et al, 2013), with previous studies describing a BH4–eNOS–mediated enhancement of vascular relaxation and diastolic function in cardiovascular disease (Baumgardt et al, 2016). The present study elucidated the therapeutic potential and mechanism of BH4 during late-stage DCM and its ability to improve mitochondrial energy metabolism via CaMKK2 and downstream CREB/PGC-1α signaling (Fig 7F).

Blood component testing suggested that BH4 treatment did not affect glucose (fasting glucose and intraperitoneal glucose tolerance test (IPGTT)) and fat metabolism (LDL, TC, and TG) in the body but rather prevented heart damage (CPK and LDH) caused by

**A**

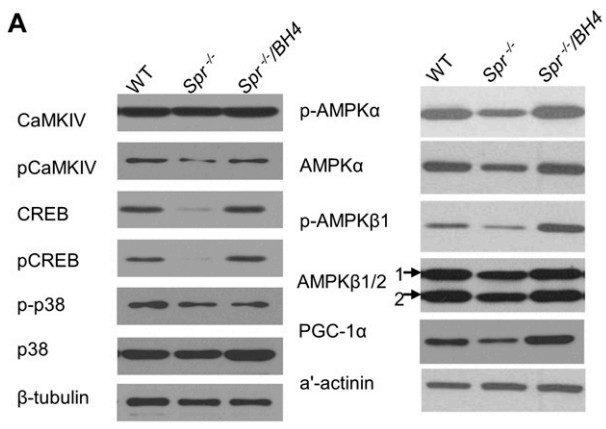

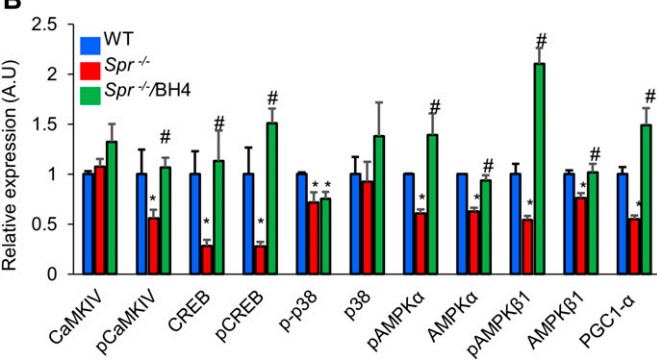

**B**

**Figure 6. BH4 regulates PGC-1α levels and AMPK phosphorylation by modulating CaMKIV/CREB signaling.**
**(A, B)** Representative Western blot (A) and quantitation (B) of total and phosphorylated signaling proteins in animal hearts. β-tubulin served as an internal standard. *$P < 0.05$ versus WT; #$P < 0.05$ versus $Spr^{-/-}$ (n = 3/group). Source data are available for this figure.

diabetes (Table 1 and Fig S1). Despite the tendency of increased lipid profiles in BH4-treated OLETF rats, recent clinical studies similarly suggested that BH4 treatment had a beneficial effect on vascular function in hypercholesterolemia patients without an effect on lipid metabolism (Nystrom et al, 2004; Cosentino et al, 2008).

We confirmed that BH4 levels are reduced in the diabetic heart. A previous study suggested that DM induces BH4 deficiency by increasing proteasome-dependent degradation of GTP cyclohydrolase 1 (GTPCH), the initial enzyme required for BH4 synthesis (Wu et al, 2016). In addition, it is possible that the diabetic condition might have promoted BH4 oxidation to its nonfunctional isoform, which competitively antagonizes BH4 function (Aviles-Herrera et al, 2017). Moreover, the high concentration of biopterin in mitochondria suggests that BH4 supplementation strongly affected mitochondrial function. In previous studies, BH4 has been reported to be compartmentalized in the cytoplasm and mitochondria. Interestingly, the concentration of BH4 was more than three times higher in mitochondria than in cardiac tissue (Shimizu et al, 2013), suggesting different roles for BH4 in each compartment. In 1972, Rembold and Buff discovered the direct effect of BH4 on the mitochondrial electron transfer chain using isolated mitochondria and sub-mitochondrial particles (Rembold & Buff, 1972). BH4 treatment directly elevated oxygen consumption and cytochrome c reduction in intact isolated rat liver mitochondria (Rembold & Buff,

1972). Consistently, BH4 treatment increased mitochondrial oxygen consumption in the absence of cellular signal intervention (Fig S7A).

Systolic dysfunction is a symptom that occurs later in DCM relative to diastolic dysfunction (Boudina & Abel, 2010), with an onset in OLETF rats at 50–60 wk, making it technically difficult to investigate (Saito et al, 2003). Extending previous studies showing the cardioprotective effect of BH4 early in DCM and other cardiovascular conditions (Baumgardt et al, 2016), we showed that BH4 restored myocardial contractility and structural deformations in late-stage rodent DCM models. In addition, we examined isolated single cardiomyocytes to demonstrate the direct influence of BH4 on heart contractility in the absence of vagus or sympathetic nerve regulation, with the results suggesting that intra-ventricular signaling pathways were targeted by BH4 in DCM.

In the present study, we used two different diabetic animal models: OLETF rats and db/db mice. OLETF is a non-insulin–dependent DM model close to human type 2 diabetes model that was developed by inbreeding. Genetically, the diabetogenic gene *Odb-1* is associated with OLETF rats. The db/db mouse was produced by inducing leptin receptor deficiency. Although, both models are widely used type 2 DM animals, we preferred to confirm the effect of BH4 in a different rodent system with a different genetic background. Another difference between the two models in the present study was that the OLETF model was a late-stage DCM model (over 48 wk), whereas the db/db mouse model was an early-stage model. In the OLETF model, BH4 treatment recovered the pathological impairment of late-stage DCM heart; in contrast, BH4 treatment prevented the progression of cardiac dysfunction in db/db mice. Therefore, we demonstrated the therapeutic potential of BH4 in both the prevention and recovery of DCM by using two different models, rat and mouse models.

In addition to cardiac contractility recovery, BH4 treatment ameliorated the detrimental DCM cardiac phenotypes, including increased proteasome activity, hypertrophied cardiac myocytes, and apoptosis in the heart of OLETF rats (Fig 2). The inhibition of proteasome activity by BH4 further suppressed pro-fibrotic factors, such as TGF-β1, collagens I and III, and monocyte chemoattractant protein-1 (Fig S3). Recently, Bailey et al (2018) similarly demonstrated that BH4 directly regulates ubiquitin-proteasome activity via eNOS-dependent S-nitrosation and that BH4 deficiency impairs the proteasome system (Bailey et al, 2018). These results suggest that BH4 suppresses pro-fibrotic signaling pathways via inhibition of proteasome activity, leading to abrogation of the observed DCM phenotypes in OLETF rats.

BH4 increased PGC-1α levels via phosphorylation of CREB and AMPK, essential regulators of cardiac energy metabolism, structural remodeling, and mitochondrial biogenesis (Daskalopoulos et al, 2016). Decreased AMPK activity in DM can lead to DCM, with a previous study describing the various therapeutic effects induced by treatment with pharmacological AMPK activators (Jeon, 2016). Interestingly, AMPK activation attenuates GTPCH degradation, thereby enhancing BH4 bioavailability (Wang et al, 2009), and BH4 supplementation increases AMPK activity and lowers blood glucose levels (Abudukadier et al, 2013), suggesting a positive regulatory feedback mechanism.

We recently reported the beneficial effects of BH4 in mitochondria in $Spr^{-/-}$ mice (Kim et al, 2019). BH4-deficient animals

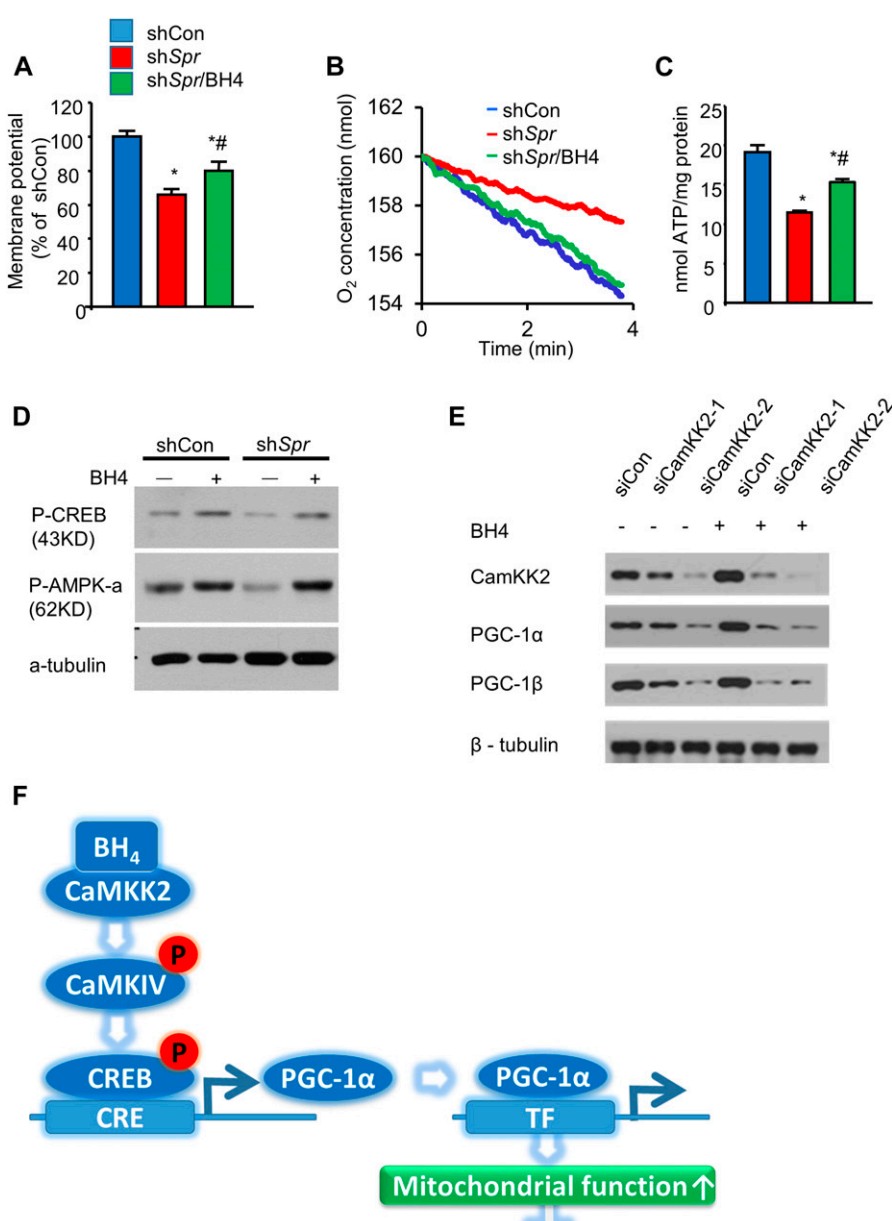

**Figure 7. BH4 regulates PGC-1α levels by modulating CaMKK2 signaling.**
**(A)** Relative mitochondrial membrane potentials. **(B, C)** Cellular oxygen consumption rates (B) and relative ATP levels (C) in indicated cell types. **(D)** Immunoblot analyses of p-CREB and p-AMPK-α in shCon, sh*Spr*, and BH4 (20 μM)-treated sh*Spr* HL-1 cells. **(E)** Representative Western blot of siCon and *Camkk2*-knockdown (siCaMKK2-1 or siCaMKK2-2) HL-1 cells in the presence or absence of BH4 (20 μM). **(F)** The proposed mechanism of BH4–CaMKK2-associated heart and mitochondrial regulation. *P < 0.05 versus shCon; #P < 0.05 versus sh*Spr* (n = 3/group).
Source data are available for this figure.

display severe cardiac dysfunction accompanied by mitochondrial defects, and we found that BH4 regulates *Pgc1a* transcription, downstream mitochondrial biogenesis, and the OXPHOS system (Kim et al, 2019). In this study, we demonstrated that both DCM condition and BH4 deficiency reduced activation of AMPK and CREB, both of which can induce the expression of *Pgc1a* and regulators of mitochondrial biogenesis. Although the reduced activation of AMPK and CREB was commonly detected in both model, the mechanism of AMPK and CREB modulation by BH4 in OLETF rats and *Spr*^−/− mice seems different. In diabetes, BH4 modulates the phosphorylation level of these proteins, without an effect on total protein level.

However, *Spr*^−/− mice showed reduced protein abundant of total and phosphorylated CREB and AMPKα. The decrease in phosphorylated AMPK and CREB protein seems to be due to the decrease in total AMPK and CREB in *Spr*^−/−.

In the present study, we identified CaMKK2, which phosphorylates and activates CaMKIV and the downstream protein CREB, as a BH4 target based on significant attenuation of CREB phosphorylation along with BH4 deficiency and restoration by BH4 supplementation. Moreover, binding assays showed that BH4 bound directly to CaMKK2 in a dose-dependent manner, and that BH4-induced increases in PGC-1α levels were abolished with *Camkk2*

knockdown. CaMKK2 plays an important role in whole-body energy homeostasis (Marcelo et al, 2016), and its transcription is decreased in aged rat hearts along with reduced levels of AMPK and PGC-1α (Barton et al, 2016). Furthermore, cardiac-specific inhibition of CaMKK2 worsens cardiac hypertrophy induced by transverse aortic binding accompanied by reduced AMPK and PGC-1α levels and mitochondrial biogenesis (Watanabe et al, 2014). However, to date, there are no known endogenous or exogenous activators of CaMKK2. As a limitation of the present study, due to lack of specific CaMKK2 inhibitor or activator, we could not determine whether other CaMKK2 modulating drugs mimic the therapeutic effects of BH4 in an in vivo model. The findings of the present study suggest CaMKK2 as a BH4 target promoting therapeutic effects associated with recovery of mitochondrial and cardiac dysfunction in DCM.

Our results demonstrated a prominent role and the underlying mechanism of BH4 in regulating heart function and cardiac mitochondrial homeostasis. These findings suggest BH4 deficiency as a possible risk factor for DCM and the therapeutic efficacy of maintaining optimal BH4 levels for the treatment of DCM with mitochondrial dysfunction.

# Materials and Methods

### Experimental animals and drug treatments

OLETF and control LETO rats (4-wk old) were purchased from Otsuka Pharmaceutical (Hayashi et al, 2003). All animals were kept in a specific pathogen–free facility with controlled temperature (20–24°C) and humidity (40–70%) on a 12-h light cycle and with access to standard laboratory chow and tap water ad libitum. All experimental procedures were approved by the Inje Medical University Animal Care and Use Committee (approval No. 2011-049) (Supplemental Data 1).

At 48-wk-old and when they began to exhibit late-stage cardiac and systolic dysfunction (Boudina & Abel, 2010), OLETF rats were divided into treated and control groups. In the treated group, 20 mg/kg/day BH4 was administered via bolus i.p. injections for 2 wk, and in the control group, rats were administered an equivalent volume of vehicle phosphate-buffered saline (Fig 1A).

At 18-wk old, male db/db (Lepr$^{db}$/J; The Charles River Japan) mice were treated with either BH4 (20 mg/kg/day via i.p. injection) or vehicle for 2 wk. For the WT control, 18-wk-old male C57BLKS/J mice were treated for 2 wk with vehicle via i.p. injection. 3-wk-old $Spr^{-/-}$ mice were treated with either BH4 (20 mg/kg/day via i.p. injection) or vehicle for 4 wk, with same-age littermate $Spr^{+/+}$ mice used as controls (Kim et al, 2019).

### One-dimensional (1D) LC–MS/MS proteomics and network analyses

We performed 1D LC–MS/MS to identify cardiac DEPs among LETO, OLETF, and OLETF/BH4 groups (Kim et al, 2012). Proteome network and enrichment analyses were conducted using STRING software (v.11.0) (Szklarczyk et al, 2019), and the protein network was visualized using Cytoscape (v.3.7.1) (Smoot et al, 2011).

### Assays

Standard procedures and methods were used for the following assays. Animal blood samples were collected for measurements of CPK, LDH, myoglobin, LDL, high-density lipoprotein, albumin, glucose, TC, blood urea nitrogen, creatinine, and TG levels. The onset of type 2 DM in the animals was determined by i.p. glucose-tolerance tests. Cardiac samples were collected from animals for assessments of morphological alterations of tissue and mitochondria via hematoxylin and eosin (H&E) and Masson's trichrome staining and EM (Dong et al, 2012). LV cardiac function was assessed by M-mode echocardiography, and electrical stimulation–induced LV cardiac cell contractility was measured by edge detection. Ca$^{2+}$ transients in beating myocytes were measured by confocal microscopy of fluorescent Fluo-4 AM–stained cardiomyocytes. BH4 levels and its oxidized species in the heart and mitochondria were analyzed by high-pressure LC. Mitochondrial function was assessed by measuring oxygen consumption rate, membrane potential, ROS, ATP production, and enzymatic OXPHOS complexes. In vitro knockdown of $Spr$ or $Camkk2$ in HL-1 cells was achieved by lentiviral transduction of shRNA or Lipofectamine transfection of siRNA, respectively. Interactions between BH4 and its candidate target kinases CaMKIV and CaMMK2 were studied by SPR analysis. Please see the online Supplementary Information for detailed experimental procedures.

### Statistical analyses

All results are expressed as the mean ± SEM. Differences between more than two groups were analyzed using one-way ANOVA followed by Bonferroni post hoc tests. All analyses were performed using GraphPad Prism 8.0 (GraphPad Software). A $P < 0.05$ was considered statistically significant.

# Supplementary Information

# Acknowledgements

Professor KI Cho, a co-author of this study, passed away in May 2020. She was a good doctor, a sincere scientist, and a very generous friend for us. We deeply mourn her passing and celebrate her life. We are all grateful for her dedication to life science and medicine. This research was supported by the Basic Research Lab Program through the National Research Foundation of Korea (NRF) funded by the Ministry of Science and ICT (MSIT) (NRF-2020R1A4A1018943) and the Basic Science Research Program through the National Research Foundation of Korea (NRF) funded by the Korea government (MSIT; 2018R1A2A3074998 and 2018R1D1A1A09081767).

### Author Contributions

HK Kim: conceptualization, data curation, formal analysis, supervision, funding acquisition, validation, investigation, and writing—original draft, review, and editing.

TH Ko: resources, data curation, formal analysis, validation, and methodology.

I-S Song: conceptualization, data curation, and formal analysis.

YJ Jeong: data curation and formal analysis.

HJ Heo: data curation and formal analysis.

SH Jeong: data curation and formal analysis.

M Kim: data curation, formal analysis, and validation.

NM Park: data curation, formal analysis, and investigation.

DY Seo: data curation, formal analysis, and investigation.

PT Kha: data curation and investigation.

S-W Kim: data curation, formal analysis, and investigation.

SR Lee: data curation, formal analysis, and investigation.

SW Cho: data curation, formal analysis, and investigation.

JC Won: data curation, formal analysis, and investigation.

JB Youm: data curation, formal analysis, and investigation.

KS Ko: formal analysis, supervision, and validation.

BD Rhee: supervision, validation, and investigation.

N Kim: formal analysis, validation, and investigation.

KI Cho: data curation, formal analysis, validation, and investigation.

I Shimizu: data curation, formal analysis, validation, investigation, and writing—original draft.

T Minamino: formal analysis, validation, investigation, and writing—review and editing.

N-C Ha: data curation, formal analysis, validation, and investigation.

YS Park: conceptualization, data curation, validation, investigation, and writing—review and editing.

B Nilius: data curation, supervision, validation, investigation, and writing—review and editing.

J Han: conceptualization, supervision, funding acquisition, validation, and writing—review and editing.

## Conflict of Interest Statement

The authors declare that they have no conflict of interest.

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
