## [Reviewer comments · Life Science Alliance]

Life Science Alliance

BH4 activates CaMKK2 and rescues the cardiomyopathic phenotype in rodent models of diabetes

Hyoung Kyu Kim, Tae Ko, In Song, Yu Jeong, Hye Jin Heo, Seung Jeong, Min Kim, Nam Mi Park, Dae Yun Seo, Pham Trong Kha, Sun-Woo Kim, Sung Ryul Lee, Sung Woo Cho, Jong Chul Won, Jae Boum Youm, Kyung Ko, Byoung Rhee, Nari Kim, Kyoung Im Cho, Ippei Shimizu, Tohru Minamino, Nam-Chul Ha, Young Park, Bernd Nilius, and Jin Han

DOI: <https://doi.org/10.26508/lsa.201900619>

Corresponding author(s): Jin Han, Cardiovascular and Metabolic Disease Center, Inje University

Review Timeline:

Submission Date:	2019-12-02
Editorial Decision:	2019-12-22
Revision Received:	2020-06-11
Editorial Decision:	2020-07-03
Revision Received:	2020-07-13
Accepted:	2020-07-14

Transaction Report:

December 22, 2019

Re: Life Science Alliance manuscript #LSA-2019-00619-T

Prof. Jin Han
Inje University
Department of Physiology, Cardiovascular and Metabolic Disease Center
Bokjiro 75, Busanjin-gu
Busan 614-735
Korea, Republic of

Dear Dr. Han,

Thank you for submitting your manuscript entitled "BH4 activates CaMKK2 and rescues the cardiomyopathic phenotype in rodent models of diabetes" to Life Science Alliance. The manuscript was assessed by expert reviewers, whose comments are appended to this letter.

As you will see, the reviewers appreciate your data. However, some of your conclusions need further support and we would thus like to invite you to submit a revised version of your manuscript to us, addressing the individual concerns raised. This seems straightforward for many of the concerns raised, and we would be happy to discuss individual revision points further with you should this be helpful.

Thank you for this interesting contribution to Life Science Alliance. We are looking forward to receiving your revised manuscript.

Sincerely,

Andrea Leibfried, PhD
Executive Editor
Life Science Alliance
Meyershofstr. 1
69117 Heidelberg, Germany
t +49 6221 8891 502
e a.leibfried@life-science-alliance.org
www.life-science-alliance.org

B. MANUSCRIPT ORGANIZATION AND FORMATTING:

Reviewer #1 (Comments to the Authors (Required)):

Kim et al. suggest that BH4 rescues the cardiomyopathic phenotype in rodent models of diabetes through CaMKK2 activation. The authors use rat diabetic cardiomyopathy (DCM) model rats (OLETF) and demonstrate that bolus i.p. treatment with BH4 improves myocardial dysfunction and

cardiac remodeling including hypertrophy, fibrosis, and apoptosis. They revealed using proteomics analysis that mitochondrial dysfunction that reducing abundances of major OXPHOS-related proteins underlie the development of DCM, and BH4 supplementation completely reverses these reductions. They also identify CaMKK2, as a critical BH4 target protein that positively regulates PGC1alpha and AMPK. Finally, they focused on the master BH4-producing enzyme, Spr, and revealed that BH4-deficient Spr^{-/-} mice displayed the significant reductions of major OXPHOS-related proteins associated with BH4 level. This study is very interesting and scientifically sound. Several concerns listed below need be revised.

1. It would be better to discuss whether BH4 participates in metabolic switch of cardiac cells or not, although almost all data are focusing on BH4-dependent regulation of OXPHOS system. Physiologically, anaerobic glycolysis will become dominant when OXPHOS system is down-regulated, but the resultant ATP content was significantly reduced. Does it mean that BH4 deficiency also lose the ability of metabolic switch in cardiomyocytes, or exceed the limit of compensation (i.e., H⁺/lactate levels are accumulated)?
2. Page 5. Abbreviation of NO and NOS should be initially described in line 1.
3. Figure 2E. It would be better to show how much percentage of myocardium undergo apoptotic phenotype rather than the total number of cells in one frame.
4. Figure 4D and d. The quantitative result of PGC1alpha protein abundances are not apparently correlated with PGC1alpha band intensity. Please replace it to clearer WB image.

Reviewer #2 (Comments to the Authors (Required)):

The authors produce evidence of the positive effect of BH4 treatment on diabetic cardiomyopathy animal models. While this effect on heart is somewhat expected from several previous reports (Liu et al., Sci Rep. 2017, 7(1):3093) and also the link of BH4 with mitochondria metabolism has been exploited (Bailey et al., Free Radic Biol Med. 2017 Mar;104:214-225. doi: 10.1016/j.freeradbiomed.2017.01.012), the manuscript reports clearcut data showing a protective effect of BH4 on different cardio-phenotypes of the investigated animal models. Interestingly, the authors propose the molecular mechanism of BH4 function, which involves the initial activation of CaMKK2.

However, few additional informations should be added.

1. Total concentration of BH4 in the tissue at basal level in OLETF and LETO rats to evaluate the claimed "BH4 deficiency".
2. Figure 4A shows only the four "cardio-relevant" pathways. We suggest to integrate with unsupervised analyses showing the non-cardio-relevant pathways, too.
3. Being BH4 a co-factor of nitric oxide synthase, how are the NOS levels in the investigated models, pre- and post-treatment?
4. please comment on why different rat and mouse models are used.

Reviewer #3 (Comments to the Authors (Required)):

The study by Kim et al investigates the effects of BH4 supplementation in diabetic cardiomyopathy. The Authors find that BH4 supplementation improves cardiac structure and function, independently of NOS activity. Indeed, BH4 is able to bind CaMKK2 that, in turn, activates CREB/AMPK/PGC1alpha pathway to trigger mitochondrial biogenesis and improve mitochondrial function.

This is an interesting study, but there are some major concerns that should be addressed:

1. It is unclear whether BH4 exerts direct effects on mitochondria or they are mediated through PGC1alpha and modulation of mitochondrial biogenesis. In the latter case, is mitochondrial content affected by BH4 supplementation? What is the significance of increasing BH4 levels in the mitochondria (Fig. 3), if BH4 acts through the cytosol-nucleus axis?
2. The idea of BH4 binding CaMKK2 and thus activating CREB is very interesting and novel. However, both CREB and AMPK protein levels are decreased in the Spr-/- mice and restored upon BH4 supplementation (Fig. 6), suggesting that BH4 may have an effect on CREB and AMPK turnover and that the mechanism of regulation of the CREB/AMPK/PGC1alpha pathway by BH4 may be different between Spr-/- mice and diabetic rats. Are total CREB levels altered in OLETF+/- BH4 rats (Fig. 4)? Do CREB and AMPK levels and phosphorylation change upon CaMKK2 silencing (Fig. 7E)?
3. BH4 supplementation causes quite a dramatic increase in LDL, total cholesterol and triglyceride levels in OLETF animals (Table 1). This needs to be addressed. The Authors should double-check whether those values are statistically different compared to the OLETF animals alone.
4. Along that line, it is advisable to include a control group treated with BH4 (i.e. LETO+BH4) to assess whether those or other parameters tested here are altered by BH4 supplementation in healthy animals.
5. Student's t-test is not the appropriate test for the comparisons carried out in this study. The Authors should include LETO+BH4 group and perform two-way ANOVA.
6. ROS formation is not affected in OLETF hearts at baseline, but only after rotenone administration. Nevertheless, MDA levels are significantly increased in OLETF hearts. Does this suggest that mitochondria are not responsible for oxidative stress in OLETF hearts? Please discuss and specify whether measurements shown in Fig. 3 were performed in tissues, cells or isolated mitochondria.
7. Mitochondrial oxygen consumption should be better characterized (measure of state 3, state 4 and maximal respiration, respiratory control ratio etc). There is an extensive description of the assay in the supplementary methods, but it is unclear what is shown in Fig. 7B.
8. The scheme shown in Fig. 7F illustrates that AMPK activation results in PGC1alpha phosphorylation and activation - this has not been demonstrated in the present study.
9. Could the Authors elaborate more on the significance of the result shown in Fig. EV2C in the context of present findings?
10. Page 11, second paragraph: The Authors conclude that the high concentration of biopterin in the mitochondria is indicative of BH4-mediated modulation of mitochondrial function. Nevertheless, the mechanism proposed in the manuscript does not support a direct effect of BH4 on mitochondria.
11. Page 12: "In the present study, we identified CaMKIV as a BH4 target..." Data presented here suggest CaMKK2 is the target.

Minor comments:

1. Page 5: "Notably, BH4 is a well-established cofactor for eNOS in DM-associated vascular disease, hypertrophic cardiac remodeling and I/R injury...". It appears that BH4 is eNOS cofactor only in disease - please rephrase.
2. Fig. EV2: Does panel B show the quantification of the zymogram shown in panel A? If so, the y-axis label is not appropriate (MMP activity was measured, not protein expression).
3. Please describe the assay shown in Fig. 5E-F in more detail.
4. Legends are missing in Fig. 7A and C.
5. Page 9: "OLETF rats showed lower levels of PGC1 alpha, the upstream regulator of p-CREB..."

The results and the scheme suggest that PGC1alpha acts downstream of CREB.

Reviewer #1 (Comments to the Authors (Required)):

Kim et al. suggest that BH4 rescues the cardiomyopathic phenotype in rodent models of diabetes through CaMKK2 activation. The authors use rat diabetic cardiomyopathy (DCM) model rats (OLETF) and demonstrate that bolus i.p. treatment with BH4 improves myocardial dysfunction and cardiac remodeling including hypertrophy, fibrosis, and apoptosis. They revealed using proteomics analysis that mitochondrial dysfunction that reducing abundances of major OXPHOS-related proteins underlie the development of DCM, and BH4 supplementation completely reverses these reductions. They also identify CaMMK2, as a critical BH4 target protein that positively regulates PGC1alpha and AMPK. Finally, they focused on the master BH4-producing enzyme, Spr, and revealed that BH4-deficient Spr^{-/-} mice displayed the significant reductions of major OXPHOS-related proteins associated with BH4 level. This study is very interesting and scientifically sound. Several concerns listed below need be revised.

1. It would be better to discuss whether BH4 participates in metabolic switch of cardiac cells or not, although almost all data are focusing on BH4-dependent regulation of OXPHOS system. Physiologically, anaerobic glycolysis will become dominant when OXPHOS system is down-regulated, but the resultant ATP content was significantly reduced. Does it mean that BH4 deficiency also lose the ability of metabolic switch in cardiomyocytes, or exceed the limit of compensation (i.e., H⁺/lactate levels are accumulated)?

Answer: Thank you for precious comment. We discovered very low ATP level in BH4 deficiency cardiac tissue and cell line. Also, AMPK phosphorylation was significantly decreased in a BH4 deficiency model. These results suggested that BH4 deficiency reduced OXPHOS capacity and glycolysis activity, resulting in severe impairment of energy metabolism, which could not be compensated in our model

2. Page 5. Abbreviation of NO and NOS should be initially described in line 1.

Answer: Thank you. We have provided the abbreviations.

3. Figure 2E. It would be better to show how much percentage of myocardium undergo apoptotic phenotype rather than the total number of cells in one frame.

Answer: Thank you for the comment. We have displayed it in a percentile manner.

4. Figure 4D and d. The quantitative result of PGC1alpha protein abundances are not apparently correlated with PGC1alpha band intensity. Please replace it to clearer WB image.

Answer: Thank you for the comment. We recalculated the protein abundances, where the intensity of PGC-1 α was divided by matched β -tubulin intensity. The quantitative result has been replaced

Reviewer #2 (Comments to the Authors (Required)):

The authors produce evidence of the positive effect of BH4 treatment on diabetic cardiomyopathy animal models. While this effect on heart is somewhat expected from several previous reports (Liu et al., Sci Rep. 2017, 7(1):3093) and also the link of BH4 with mitochondria metabolism has been exploited (Bailey et al., Free Radic Biol Med. 2017 Mar;104:214-225. doi: 10.1016/j.freeradbiomed.2017.01.012), the manuscript reports clearcut data showing a protective effect of BH4 on different cardio-phenotypes of the investigated animal models. Interestingly, the authors propose the molecular mechanism of BH4 function, which involves the initial activation of CaMKK2. However, few additional informations should be added.

1. Total concentration of BH4 in the tissue at basal level in OLETF and LETO rats to evaluate the claimed "BH4 deficiency".

Answer: Thank you for the comment. We have added the BH4 concentration, besides the total biopterin level.

2. Figure 4A shows only the four "cardio-relevant" pathways. We suggest to integrate with unsupervised analyses showing the non-cardio-relevant pathways, too.

Answer: Thank you for the comment. We added another network, which includes 'non-cardio-relevant pathways, such as proteasome and glutathione metabolism pathways. The new network has been added as Figure S4. We have also described the additional information in the result section as below.

. In addition to the four major cardiac regulatory pathways, functional enrichment associated with 'glutathione metabolism' or 'proteasome' was also identified. These pathways appear to be associated with oxidative stress and pathological remodeling, respectively, in DCM (Figure S4).

3. Being BH4 a co-factor of nitric oxide synthase, how are the NOS levels in the investigated models, pre- and post-treatment?

Answer: Thank you for the comment. We tested the level of eNOS and phosphor-eNOS in the heart of LETO, OLETF, and OLET/BH4 rats. The ratio of p-eNOS/eNOS in the BH4 treatment group tended to increase, but there was no statistical significance in the ANOVA test.

We have described the additional information in the result section as below.

To confirm the change in eNOS-mediated NO signal by BH4 treatment, we tested the level of eNOS and phosphor-eNOS in the heart of LETO, OLETF, and OLET/BH4 rats. The ratio of p-eNOS/eNOS in the BH4 treatment group tended to increase, but there was no statistical significance (Fig S2F).

4. please comment on why different rat and mouse models are used.

Answer: Thank you for the comment. In the present study, we used two different diabetic animal models: OLETF rats and db/db mice. OLETF is a non-insulin-dependent DM model close to human type 2 diabetes model that was developed by inbreeding. Genetically, the diabetogenic gene *Odb-1* is associated with OLETF rats. The db/db mouse was produced by inducing leptin receptor deficiency. Although, both models are widely used type 2 DM animals, we preferred to confirm the effect of BH4 in a different rodent system with a different genetic background. Another difference between the two models in the present study was that the OLETF model was a late-stage DCM model (over 48 weeks), whereas the db/db mouse model was an early-stage model. In the OLETF model, BH4 treatment recovered the pathological impairment of late-stage DCM heart; in contrast, BH4 treatment prevented the progression of cardiac dysfunction in db/db mice. Therefore, we demonstrated the therapeutic

potential of BH4 in both the prevention and recovery of DCM by using two different models, rat and mouse models.

Reviewer #3 (Comments to the Authors (Required)):

The study by Kim et al investigates the effects of BH4 supplementation in diabetic cardiomyopathy. The Authors find that BH4 supplementation improves cardiac structure and function, independently of NOS activity. Indeed, BH4 is able to bind CaMKK2 that, in turn, activates CREB/AMPK/PGC1alpha pathway to trigger mitochondrial biogenesis and improve mitochondrial function.

This is an interesting study, but there are some major concerns that should be addressed:

1. It is unclear whether BH4 exerts direct effects on mitochondria or they are mediated through PGC1alpha and modulation of mitochondrial biogenesis. In the latter case, is mitochondrial content affected by BH4 supplementation? What is the significance of increasing BH4 levels in the mitochondria (Fig. 3), if BH4 acts through the cytosol-nucleus axis?

Answer: Thank you for important comment. BH4 has been reported to be compartmentalized in the cytoplasm and mitochondria. Interestingly, the concentration of BH4 was more than 3 times higher in mitochondria than in cardiac tissue ¹, suggesting different roles for BH4 in each compartment. In 1972, Rembold and Buff discovered the direct effect of BH4 on the mitochondrial electron transfer chain using isolated mitochondria and sub-mitochondrial particles². BH4 treatment directly elevated oxygen consumption and cytochrome c reduction in intact isolated rat liver mitochondria.

To confirm this direct effect of BH4 on cardiac mitochondria, we analyzed the effect of BH4 on mitochondrial oxygen consumption in purely isolated mitochondria of mice. In the presence of BH4, mitochondrial oxygen consumption increased in both state 4 and state 3 without cytosolic influence.

In addition, an increase in mitochondrial biosynthesis by BH4 was confirmed in HL-1 cells. The mitochondrial mass was measured by acridine orange 10-nonyl bromide (NAO; Invitrogen), which binds to the mitochondrial membrane phospholipid, cardiolipin. BH4 treatment also increased the protein level of PGC-1 α , mt-TFA, and NRF1, which are key regulators of mitochondrial biogenesis, in treated WT mice.

1. Oxygen consumption rate in isolated cardiac mitochondria of mice.

2. BH4 treatment increased mitochondrial mass in HL-1 cells

3. BH4 treatment increased the expression of mitochondria biogenesis-related proteins

Based on these result, we proposed that BH4 enhanced mitochondria function both directly and indirectly. We have added these results and interpretation in the revised version of the manuscript.

2. The idea of BH4 binding CaMKK2 and thus activating CREB is very interesting and novel. However, both CREB and AMPK protein levels are decreased in the Spr^{-/-} mice and restored upon BH4 supplementation (Fig. 6), suggesting that BH4 may have an effect on CREB and AMPK turnover and that the mechanism of regulation of the CREB/AMPK/PGC1alpha pathway by BH4 may be different between Spr^{-/-} mice and diabetic rats. Are total CREB levels altered in OLETF^{+/-} BH4 rats (Fig. 4)? Do CREB and AMPK levels and phosphorylation change upon CaMKK2 silencing (Fig. 7E)?

Answer: Thank you for the valuable comment. We tested the protein level of CREB in the LETO, OLETF, and OLETF/BH4 rats. There was no significant different among the groups. We tested the level of total and phosphorylation level of AMPK protein in CaMKK2-silenced HL-1 cells. The level of total AMPK was not altered by CamKK2 knockdown, while the phosphorylation of AMPK was decreased by CamKK2 knockdown. However, we could not confirm the phosphorylation change of CREB, due to limitation of materials under COVID situation. We would like to ask reviewer's understanding.

1. Total CREB level in OLETF (western check)

2. CaMKK2 knockdown effect on total and phosphorylated AMPK level

3. BH4 supplementation causes quite a dramatic increase in LDL, total cholesterol and triglyceride levels in OLETF animals (Table 1). This needs to be addressed. The Authors should double-check whether those values are statistically different compared to the OLETF animals alone.

Answer: Thank you for the valuable comment. We checked the mean difference between the OLETF and OLETF+BH4 groups. Although the mean values of LDL, TC, and TG were higher in the OLETF+BH4 group than in the OLETF group, the difference was not statistically significant. Even though there was no statistical significance, the reviewer's question of 'whether BH4 affect lipid metabolism directly or not' is important. Therefore, we have added the reviewer's consideration in the discussion section as below

Blood component testing suggested that BH4 treatment did not affect glucose (fasting glucose and IPGTT) and fat metabolism (LDL, TC and TG) in the body but rather prevented heart damage (CPK and LDH) caused by diabetes (Table 1 and Fig S1). Despite the tendency of increased lipid profiles in BH4-treated OLETF rats, recent clinical studies similarly suggested that BH4 treatment had a beneficial effect on vascular function in hypercholesterolemia patients without an effect on lipid metabolism^{3,4}.

4. Along that line, it is advisable to include a control group treated with BH4 (i.e. LETO+BH4) to assess whether those or other parameters tested here are altered by BH4 supplementation in healthy animals.

- Answer: Thank you for the valuable comment. We agree your suggestion; however, we could not perform those experiments due to the limitation of materials. Further, it will take about a year to get age-matched LETO animals to perform new experiments. Instead, we added new data that BH4 treatment in healthy normal mouse still enhanced the protein level of PGC-1 α and related proteins including NRF-1 and mtTFA.

5. Student's t-test is not the appropriate test for the comparisons carried out in this study. The Authors should include LETO+BH4 group and perform two-way ANOVA.

Answer : Thank you for the comment. As our response above, we could not perform these experiments due to the limitation of materials. Further, it will take about a year to get age-matched LETO animals to perform new experiments. We recalculated our result by one-way

ANOVA for mean comparison among LETO, OLETF, and OLETF+BH4.

6. ROS formation is not affected in OLETF hearts at baseline, but only after rotenone administration. Nevertheless, MDA levels are significantly increased in OLETF hearts. Does this suggest that mitochondria are not responsible for oxidative stress in OLETF hearts? Please discuss and specify whether measurements shown in Fig. 3 were performed in tissues, cells or isolated mitochondria.

Answer: Thank you for the comment. We performed the experiment in isolated mitochondria. At the basal level, where there was only mitochondria without substrate, mitochondria produced very low levels of ROS. However, in the presence of a substrate and the complex I inhibitor, rotenone, OLETF cardiac mitochondria produced significantly high ROS levels compared with those in the others. Since the diabetic heart had been exposed to metabolic stress and could not use glucose but overused free fatty acid, the sufficient substrate and mitochondria stress condition may be close to the *in vivo* condition of the heart. MDA level reflected the accumulated *in vivo* oxidative stress. Thus, the result suggests that increased ROS production under stress condition in mitochondria resulted in higher lipid peroxidation (high MDA level)

7. Mitochondrial oxygen consumption should be better characterized (measure of state 3, state 4 and maximal respiration, respiratory control ratio etc). There is an extensive description of the assay in the supplementary methods, but it is unclear what is shown in Fig. 7B.

Answer: The oxygen consumption result in Figure 7B is the measured cellular oxygen consumption in the HL-1 cardiac cell line. We measured the total cellular oxygen consumption rate in the shCon, shSpr, and shSpr/BH4 groups as follows (we have added this method to the online information)

Cellular Oxygen Consumption

Cultured HL-1 cells were harvested and washed twice with PBS and then resuspended in fresh complete RPMI medium at a density of 2.5×10^7 cells/mL. Each experiment measured oxygen consumption by 400 μ L of resuspended cells in a 600- μ L air-saturated chamber surrounded by a water-filled chamber maintained at 37 °C, using fiber-optic oxygen electrode (Instech). The oxygen concentration in fully oxygenated medium and cell cultures was taken to be 200 nmol/mL (200 μ M). Oxygen consumption rates were calculated in nmoles per million cells per second.

In the revised version of the manuscript, we analyzed the effect of BH4 on mitochondrial oxygen consumption in purely isolated mitochondria of mice to confirm this direct effect of BH4 on cardiac mitochondria. In the presence of BH4, mitochondrial oxygen consumption

increased in both state 4 and state 3.

8. The scheme shown in Fig. 7F illustrates that AMPK activation results in PGC1alpha phosphorylation and activation - this has not been demonstrated in the present study.

Answer: Thank you. We modified the scheme with only our confirmed data.

F

9. Could the Authors elaborate more on the significance of the result shown in Fig. EV2C in the context of present findings?

Answer: Thank you for the constructive advice. As the preventive effect of BH₄ on cardiac hypertrophy and fibrosis, we found that BH₄ treatment also modulated the proteasome activity in the heart through proteomic analysis. It was suggested that the diabetic condition highly increased proteasome activity. We tested the effect of BH₄ in the regulation of

proteasome activity in the heart tissue. As shown in Figure S3. PROMMP-2 and -9 activity was significantly increased in OLETF rats, which was reduced by BH4 treatment. In the proteasome activity assay (Fig S3C), 26s proteasome inhibitor, a lactacystin, treatment abolished the proteasome-inhibiting effect of BH4, suggesting that BH4 inhibits proteasome activity and detrimental cardiac remodeling. Recently, Bailey et al. similarly demonstrated that BH4 directly regulates ubiquitin-proteasome activity via eNOS-dependent S-nitrosation and that BH4 deficiency impairs the proteasome system. We have added the detailed description of the presented results in the discussion section, as below.

In addition to cardiac contractility recovery, BH4 treatment ameliorated the detrimental DCM cardiac phenotypes, including increased proteasome activity, hypertrophied cardiac myocytes, and apoptosis in the heart of OLETF rats (Fig 2). The inhibition of proteasome activity by BH4 further suppressed pro-fibrotic factors, such as TGF- β 1, collagens I and III, and MCP-1 (Fig S3). Recently, Bailey *et al.* similarly demonstrated that BH4 directly regulates ubiquitin-proteasome activity via eNOS-dependent S-nitrosation and that BH4 deficiency impairs the proteasome system⁵. These results suggest that BH4 suppresses pro-fibrotic signaling pathways via inhibition of proteasome activity, leading to abrogation of the observed DCM phenotypes in OLETF rats.

10. Page 11, second paragraph: The Authors conclude that the high concentration of biopterin in the mitochondria is indicative of BH4-mediated modulation of mitochondrial function. Nevertheless, the mechanism proposed in the manuscript does not support a direct effect of BH4 on mitochondria.

Answer: Thank you for the important comment. We discussed carefully the possibility of BH4 effect on mitochondria using previous studies. In the revised version of the manuscript, we analyzed the effect of BH4 on mitochondrial oxygen consumption in purely isolated mitochondria of mice to confirm this direct effect of BH4 on cardiac mitochondria. In the presence of BH4, mitochondrial oxygen consumption increased in both state 4 and state 3.

11. Page 12: "In the present study, we identified **CaMKIV** as a BH4 target..." Data presented here suggest CaMKK2 is the target.

Answer: We have corrected this as below

In the present study, we identified **CaMKK2**, which phosphorylates and activates **CaMKIV** and the downstream protein **CREB**, as a BH4 target based on significant attenuation of **CREB** phosphorylation along with BH4 deficiency and restoration by BH4 supplementation.

Minor comments:

1. Page 5: "Notably, BH4 is a well-established cofactor for eNOS in DM-associated vascular disease, hypertrophic cardiac remodeling and I/R injury...". It appears that BH4 is eNOS cofactor only in disease - please rephrase.

Answer: We corrected this as below

Notably, BH4 is a well-established cofactor for endothelial nitric oxide synthase (eNOS/NOS3), regulating vascular and cardiac function. The BH4-eNOS uncoupling is associated with vascular disease, hypertrophic cardiac remodeling, and ischemia–reperfusion injury.

2. Fig. EV2: Does panel B show the quantification of the zymogram shown in panel A? If so, the y-axis label is not appropriate (MMP activity was measured, not protein expression).

Answer: Thank you. We corrected it

3. Please describe the assay shown in Fig. 5E-F in more detail.

Answer: We have described it in more detail as below

To test the direct binding of BH4 and CaMKIV or CaMKK2, we performed a computational protein-ligand docking simulation and surface plasmon resonance (SPR) binding assay. Computational protein-ligand docking simulation suggested a probable binding of BH4 to CaMKIV (PDB: 2W4O) or indicated CaMKK2 (PDB:6CMJ) domain residues by

polar, hydrophobic, and hydrogen bond interactions (Fig 5 A-D). The SPR assay was performed to validate the docking simulation results of BH4 binding to CaMKIV or CaMKK2, which phosphorylates AMPK (Racioppi & Means, 2012). The SPR analysis revealed that CaMKK2 directly and dose-dependently bound to BH4 (Fig 5F); however, CaMKIV was not the direct binding target of BH4 (Fig 5E).

4. Legends are missing in Fig. 7A and C.

Answer: Thank you. We have added the legends.

Figure 7 - BH4 regulates PGC-1 α levels by modulating CaMKK2 signaling. (A) Relative mitochondrial membrane potentials. (B, C) Cellular oxygen-consumption rates (B) and relative ATP levels (C) in indicated cell types.

5. Page 9: "OLETF rats showed lower levels of PGC1 alpha, the upstream regulator of p-CREB..." The results and the scheme suggest that PGC1alpha acts downstream of CREB.

Answer: Thank you. You are right. We have correct this mistake as below

OLETF rats showed lower levels of p-CREB, the upstream regulator of PGC-1 α (Fig 4D),

1. Shimizu S, Ishibashi M, Kumagai S, Wajima T, Hiroi T, Kurihara T *et al.* Decreased cardiac mitochondrial tetrahydrobiopterin in a rat model of pressure overload. *Int J Mol Med* 2013;**31**:589-596.
2. Rembold H, Buff K. Tetrahydrobiopterin, a cofactor in mitochondrial electron transfer. Effect of tetrahydropterins on intact rat-liver mitochondria. *Eur J Biochem* 1972;**28**:579-585.
3. Cosentino F, Hurlimann D, Delli Gatti C, Chenevard R, Blau N, Alp NJ *et al.* Chronic treatment with tetrahydrobiopterin reverses endothelial dysfunction and oxidative stress in hypercholesterolaemia. *Heart* 2008;**94**:487-492.
4. Nystrom T, Nygren A, Sjöholm A. Tetrahydrobiopterin increases insulin sensitivity in patients with type 2 diabetes and coronary heart disease. *American journal of physiology Endocrinology and metabolism* 2004;**287**:E919-925.
5. Bailey J, Davis S, Shaw A, Diotallevi M, Fischer R, Benson MA *et al.* Tetrahydrobiopterin modulates ubiquitin conjugation to UBC13/UBE2N and proteasome activity by S-nitrosation. *Sci Rep* 2018;**8**:14310.

July 3, 2020

RE: Life Science Alliance Manuscript #LSA-2019-00619-TR

Prof. Jin Han
Inje University
Department of Physiology, Cardiovascular and Metabolic Disease Center
Bokjiro 75, Busanjin-gu
Busan 614-735
Korea, Republic of

Dear Dr. Han,

Thank you for submitting your revised manuscript entitled "BH4 activates CaMKK2 and rescues the cardiomyopathic phenotype in rodent models of diabetes". We would be happy to publish your paper in Life Science Alliance pending final revisions necessary to meet our formatting guidelines.

With respect to the remaining minor issues of referee #3 we ask you to revisit the manuscript to improve presentation and discussion of the findings, introducing caveats where appropriate and toning down claims where needed. Please also adjust the schematic in Fig XY and take care of these formatting requirements:

- please check that the author names in the manuscript and our system match (author Pham Trong Kha in manuscript text, but in system Trong Kha Pham)
- please have corresponding authors add their ORCID ID-you should have received instructions on how to do so
- please add a callout to Figure S2D in the manuscript text
- please add the supplementary figure legends to the main manuscript text and upload the supplementary figures without the legends below the figures
- please add the panel Fig 1J to the Figure 1 Legend
- For Figure S2, you have panels A-D rather than A-C. Please fix.
- Please add panels A & B to figure S5

A. FINAL FILES:

B. MANUSCRIPT ORGANIZATION AND FORMATTING:

Sincerely,

Reilly Lorenz

Reviewer #3 (Comments to the Authors (Required)):

This is a revised version so I will only concentrate on the points I already made. The Authors performed additional experiments and partially addressed the concerns I raised during the first round of revision. In my reading there are still a few issues that need to be resolved:

1. The Authors performed respiration experiments in mitochondria isolated from mouse hearts (Figure S7A), but the incubation conditions are not specified either in the legend or in the methods section. At face value, the data show that BH4 supplementation increases respiration both in state 4 (no ADP) and in state 3 (with ADP; and phosphate, I guess; I assume that there are substrates here too, or else the experiment makes little sense and in this case it should be repeated properly). If so, while this supports the direct effect of BH4 on mitochondria, the fact that the rate of oxygen consumption is the same in state 3 and state 4, regardless of the presence of BH4, raises some concerns regarding the quality of the mitochondrial preparation (the respiratory control index appears to be close to 1, i.e. no stimulation by ADP). Moreover, rates of oxygen consumption are quite low compared to what was previously reported for cardiac mitochondria. The Authors should check for the integrity and quality of their preparation. Legend for the graph shown in Suppl. Fig. 7A is missing.
2. The mechanism of AMPK and CREB modulation by BH4 in OLETF rats and Spr^{-/-} mice still seems different: while in diabetes it appears that BH4 modulates the phosphorylation level of these proteins, in Spr^{-/-} mice it seems that the reduced phosphorylation is simply the consequence of lower total protein abundance (indeed, levels of phosphorylated AMPK and CREB shown in Fig. 6 were normalized to beta-tubulin rather than to the total AMPK or CREB levels. In relation to this, please rephrase the sentence on page 10 regarding phosphorylation levels). The Authors could introduce a limitation paragraph in the Discussion to address this discrepancy. Also, please include quantification for the representative western blots shown in Suppl. Fig. 8 and a housekeeping protein for the blots shown in panel B.
3. Regarding the increase in oxygen consumption in shSpr/BH4 cells, it needs to be stated in the text (page 11) that what is shown in Fig. 7B reflects total cellular oxygen consumption and is not necessarily related only to mitochondrial respiration (importantly, it was not assessed whether this is coupled to an increase in ATP synthesis).
4. Page 11: "In addition, CaMKK2 knockdown in HL-1 cells via small-interfering (si)RNA decreased the level of PGC-1 α , -1 β , and AMPK phosphorylation, which further abolished BH4-induced increases in PGC-1 α levels (Fig 7E and Fig S8)." Further respect to what?
5. Regarding the scheme shown in Fig. 7F, the Authors removed the relationship between AMPK and PGC1alpha phosphorylation. Nevertheless, I still find the scheme a bit misleading, since PGC1alpha phosphorylation (still depicted) was not assessed here and is not necessarily required

for mitochondrial biogenesis to occur.

Executive Editor
Life Science Alliance

Subject: LSA-2019-00619-T

Dear Editors,

We are very happy to publish our paper in Life Science Alliance. We reformatted our final version of manuscript according to the journal's guideline. Also we carefully addressed the issues of referee #3. Thanks again to Referee #3 for your valuable and precise comments.

Thank you

Sincerely yours,

Jin Han, MD, PhD

Professor

National Research Laboratory for Mitochondrial Signaling, Department of Physiology,
College of Medicine, Department of Health Sciences and Technology, BK21 Plus Project
Team, Smart Marine Therapeutics Center, Cardiovascular and Metabolic Disease Center, Inje
University

Bokji-ro 75, Busanjin-gu, Busan, 47392, Korea

Tel: +82-51-890-6727

Fax: +82-51-891-8748

Email: phyhanj@inje.ac.kr

Response to Editorial office and Reviewer's Comment

1. Editorial Request

-please check that the author names in the manuscript and our system match (author Pham Trong Kha in manuscript text, but in system Trong Kha Pham)

A: We corrected the author's name in the manuscript and system as "Pham Trong Kha"

-please have corresponding authors add their ORCID ID-you should have received instructions on how to do so

A: We added ORCID ID of corresponding author

-please add a callout to Figure S2D in the manuscript text

A: The label of S2D was mistaken. We fixed the label of S2D as S2C. Fig S2C was mentioned in the result section as follow.

a similar effect was observed in 20-week-old *db/db* model hearts via two-dimensional (2D) M-mode echocardiography (Fig S2A–C), which revealed significantly improved cardiac contractility.

-please add the supplementary figure legends to the main manuscript text and upload the supplementary figures without the legends below the figures

A: We added the supplementary figure legends to the main manuscript text

-please add the panel Fig 1J to the Figure 1 Legend

A: We added the legend of Fig 1J as follow

(J) Velocity of shortening and Re-lengthening of isolated LV cardiomyocytes ($n = 15/\text{group}$)

-For Figure S2, you have panels A-D rather than A-C. Please fix.

A: We fixed the label of S2D as S2C.

-Please add panels A & B to figure S5

A: We added panels A & B to figure S5

A

B

2. Response to reviewer's comment

Reviewer #3 (Comments to the Authors (Required)):

This is a revised version so I will only concentrate on the points I already made. The Authors performed additional experiments and partially addressed the concerns I raised during the first round of revision. In my reading there are still a few issues that need to be resolved:

1. The Authors performed respiration experiments in mitochondria isolated from mouse hearts (Figure S7A), but the incubation conditions are not specified either in the legend or in the methods section. At face value, the data show that BH4 supplementation increases respiration both in state 4 (no ADP) and in state 3 (with ADP; and phosphate, I guess; I assume that there are substrates here too, or else the experiment makes little sense and in this case it should be repeated properly). If so, while this supports the direct effect of BH4 on mitochondria, the fact that the rate of oxygen consumption is the same in state 3 and state 4, regardless of the presence of BH4, raises some concerns regarding the quality of the mitochondrial preparation (the respiratory control index appears to be close to 1, i.e. no stimulation by ADP). Moreover, rates of oxygen consumption are quite low compared to what was previously reported for cardiac mitochondria. The Authors should check for the integrity and quality of their preparation. Legend for the graph shown in Suppl. Fig. 7A is missing.

A: Thank you for constructive comments.

(1) Final concentration of 20 μM of BH4 was added into the OCR assay chamber together with isolated mitochondria without pre-incubation to test direct effect of BH4 on mitochondrial OCR. We added this incubation condition of BH4 in the result part as below

To confirm this direct effect of BH4 on cardiac mitochondria, we analyzed the effect of BH4 on mitochondrial oxygen consumption in purely isolated mitochondria of mice. Final concentration of 20 μM of BH4 was added into the OCR assay chamber together with isolated mitochondria without pre-incubation. After a stabilization period of 5 minutes, glutamate/malate and ADP were sequentially added to measure mitochondrial oxygen consumption.

(2) We retested the effect of BH4 on mitochondrial OCR with proper RCR value. Also the unit of OCR was reevaluated.

2. The mechanism of AMPK and CREB modulation by BH4 in OLETF rats and *Spr*^{-/-} mice still seems different: while in diabetes it appears that BH4 modulates the phosphorylation level of these proteins, in *Spr*^{-/-} mice it seems that the reduced phosphorylation is simply the consequence of lower total protein abundance (indeed, levels of phosphorylated AMPK and CREB shown in Fig. 6 were normalized to beta-tubulin rather than to the total AMPK or CREB levels. In relation to this, please rephrase the sentence on page 10 regarding phosphorylation levels).

The Authors could introduce a limitation paragraph in the Discussion to address this discrepancy. Also, please include quantification for the representative western blots shown in Suppl. Fig. 8 and a housekeeping protein for the blots shown in panel B.

A: Thank you for the important comment. We rephrase the sentence on page 10 and also address this discrepancy in the discussion part. We included quantification for the representative western blots shown in Suppl. Fig. 8 and a housekeeping protein for the blots shown in panel B.

On page 10 result

Importantly, *Spr* knockdown also reduced protein level of total and phosphorylated CREB and AMPK- α , which was subsequently enhanced by BH4 supplementation (Fig 7D).

On page 14 discussion

Although, the reduced activation of AMPK and CREB was commonly detected in both model, while, the mechanism of AMPK and CREB modulation by BH4 in OLETF rats and *Spr*^{-/-} mice seems different. In diabetes, BH4 modulates the phosphorylation level of these proteins, without an effect on total protein level. While *Spr*^{-/-} mice showed reduced protein abundant of total and phosphorylated CREB and AMPK α . The decrease in phosphorylated AMPK and CREB protein seems to be due to the decrease in total AMPK and CREB in *Spr*^{-/-}.

Figure S8B

3. Regarding the increase in oxygen consumption in shSpr/BH4 cells, it needs to be stated in the text (page 11) that what is shown in Fig. 7B reflects total cellular oxygen consumption and is not necessarily related only to mitochondrial respiration (importantly, it was not assessed whether this is coupled to an increase in ATP synthesis).

A: Thank you for the important comment. We indicated OCR in Fig. 7B reflected total cellular oxygen consumption in the result section.

On page 10 result

Depletion of the *Spr* gene depolarized mitochondrial inner-membrane potentials, reduced total cellular oxygen consumption, and decreased ATP levels in HL-1 cells (Fig 7A–C),

4. Page 11: "In addition, CaMKK2 knockdown in HL-1 cells via small-interfering (si)RNA decreased the level of PGC-1 α , -1 β , and AMPK phosphorylation, which further abolished BH4-induced increases in PGC-1 α levels (Fig 7E and Fig S8)." Further respect to what?

A: Thank you for the valuable comment. We removed "further" for clarity and rephrase the sentence as follow.

On page

In addition, *CaMKK2* knockdown in HL-1 cells via small-interfering (si)RNA decreased the level of PGC-1 α , -1 β , and AMPK phosphorylation, which abolished BH4-induced increases in PGC-1 α levels

5. Regarding the scheme shown in Fig. 7F, the Authors removed the relationship between AMPK and PGC1alpha phosphorylation. Nevertheless, I still find the scheme a bit misleading, since PGC1alpha phosphorylation (still depicted) was not assessed here and is not necessarily required for mitochondrial biogenesis to occur.

A: Thank you for the comment. We edited the Fig. 7F as follow your comment. The phosphorylation of PGC-1 α was removed in the revised version.

July 14, 2020

RE: Life Science Alliance Manuscript #LSA-2019-00619-TRR

Dr. Jin Han
Cardiovascular and Metabolic Disease Center, Inje University
Bokji-ro 75
Busan
Korea (South), Republic of

Dear Dr. Han,

Thank you for submitting your Research Article entitled "BH4 activates CaMKK2 and rescues the cardiomyopathic phenotype in rodent models of diabetes". It is a pleasure to let you know that your manuscript is now accepted for publication in Life Science Alliance. Congratulations on this interesting work.

DISTRIBUTION OF MATERIALS:

Again, congratulations on a very nice paper. I hope you found the review process to be constructive and are pleased with how the manuscript was handled editorially. We look forward to future exciting submissions from your lab.

Sincerely,

Reilly Lorenz
Editorial Office Life Science Alliance
Meyerhofstr. 1
69117 Heidelberg, Germany
t +49 6221 8891 414
e contact@life-science-alliance.org
www.life-science-alliance.org